# Visualization of Medical Record with 3D Human Body Models

**DOI:** 10.3390/healthcare13192393

**Published:** 2025-09-23

**Authors:** Tz-Jie Liu, Chia-Yi Lai, Yi-Cheng Chiang

**Affiliations:** 1Department of Information Management, National Chung-Cheng University, Chiayi 621301, Taiwan; jie@stm.org.tw (T.-J.L.); chiayi@mis.ccu.edu.tw (C.-Y.L.); 2Center of Health Management, St. Martin De Porres Hospital, Chiayi 600044, Taiwan; 3Taichung Tzu-Chi Hospital, The Buddhist Tzu Chi Medical Foundation, Taichung 427213, Taiwan

**Keywords:** smart healthcare, electronic medical record system, 3D technology, 3D human model, information visualization

## Abstract

**Background/Objectives:** With the rapid development of smart healthcare, medical records have shifted from a disease-centered to a patient-centered approach. However, traditional formats, such as narratives and tables, often make it challenging for physicians to quickly grasp a patient’s condition within a limited timeframe, potentially leading to diagnostic errors and a decline in the quality of care. Recently, advances in information visualization and 3D technology have led many medical institutions to employ charts and graphs or use 3D simulations of organs to support clinical practice and education. However, few have integrated 3D models into medical records for use during physician consultations. **Methods:** This study presents the development and evaluation of a novel web-based 3D EMR system that integrates real-time ICD-10 diagnostic code mapping with interactive 3D human body models, enabling physicians to visualize patient-specific anatomical and diagnostic information in a dynamic and context-aware manner. **Results:** We employed the System Usability Scale (SUS) to evaluate the system’s usability, conducting a satisfaction survey. Results from the survey indicate that participants rated the system highly in terms of ease of use, satisfaction, and efficiency, with an average SUS score of 70.42, reflecting usability between moderate and good. Comparative evaluations and future expansion plans are also discussed. **Conclusions**: These findings demonstrate that integrating a 3D human model into the medical record retrieval process significantly improves visualization and interactivity, meeting the needs of healthcare professionals and enhancing both their efficiency and patient satisfaction.

## 1. Introduction

In the context of smart healthcare, the rapid digitalization of clinical systems has catalyzed a shift from disease-centered data structures to patient-centered visual interaction models. Traditional EMRs, typically presented in text or table formats, are often unintuitive and cognitively demanding, impeding physicians’ ability to make timely decisions—especially in complex or multidisciplinary cases [1,2]. To address this, information visualization (InfoVis) has emerged as a critical tool for reducing cognitive load and facilitating insight discovery from large-scale medical data [3,4]. However, most existing InfoVis approaches rely on static 2D representations (e.g., color-coded interfaces and timelines), which lack spatial realism, interactivity, and anatomical fidelity [4,5,6].

Recent advancements in 3D modeling and WebGL rendering have expanded the possibilities for immersive and anatomically accurate EMR visualization. Three-dimensional anatomical models have proven useful in radiology, surgical planning, and medical education [7,8,9,10,11], yet they are seldom embedded directly into the EMR interfaces used in daily clinical workflows. Some earlier efforts, such as Jin’s 2D-body-based visualization system [9,11], made important strides in mapping symptom locations to visual metaphors, but they lacked internal anatomical depth, multi-system integration, and were limited to static viewing angles. However, these traditional approaches have notable limitations: they often lack intuitiveness and interactivity, making it difficult for physicians to quickly and accurately retrieve patient histories, which can hinder the understanding of diagnostic records.

In clinical practice, visualized medical data could allow healthcare professionals to intuitively and efficiently comprehend, differentiate, and manage patient information, facilitating rapid retrieval of useful insights from extensive medical records. This is particularly valuable for diseases with physical symptoms, such as angina, mitral valve prolapse, anterior cruciate ligament rupture, and gastric cancer [6,12,13].

With the rapid evolution of web technologies, 3D visualization tools have become increasingly prevalent in web applications, significantly enhancing interactivity and visual effects while providing users with immersive experiences. In healthcare, 3D web tools effectively present complex medical information, such as human anatomy and pathological changes, thus improving the clarity and comprehension of medical data. Physicians can use this technology for precise diagnoses, leveraging detailed 3D models to better understand patients’ pathological conditions, thereby enhancing diagnostic accuracy and efficiency. Furthermore, 3D medical models not only improve physician–patient communication but also facilitate more accurate treatment planning, elevating the quality of medical information presentation. Given that many hospitals now operate hospital information systems on web platforms, this study integrates 3D human models into a physician’s medical record entry system (referred to as the doctor entry system). We introduce a web-based 3D human model into the medical records of individual patients. Utilizing WebGL technology, we have developed a professional, scalable, and fully functional 3D web-based human model system [14]. This system dynamically generates models based on medical records, enabling real-time querying and facilitating seamless communication between physicians and patients. The interface is user-friendly, providing a streamlined way to present anatomical and disease information, thus helping physicians quickly assess patient health conditions. By visualizing medical records, the system enhances the immersive experience during consultations, offering substantial benefits for medical applications. This approach also holds potential to reduce medical costs and opens new possibilities in the healthcare field. Ultimately, this system aims to improve physician–patient communication, foster patient trust in medical decisions, and support patients in gaining a better understanding of their health status [15].

In addition to proposing this system, we aim to assess its usability and practicality in clinical settings. Our system was evaluated by 34 healthcare professionals using the System Usability Scale (SUS). Results demonstrate that the system facilitates effective exploration of patient data and offers a satisfying user experience. In response to challenges like system performance and visualizing non-physical conditions, we also explore performance optimization strategies (e.g., LOD control, caching), and future integration of ontology-based symptom mapping. Through this study, we contribute not only a technically viable prototype but also present a practical clinical framework for embedding interactive 3D anatomical models into EMRs—bridging the gap between abstract diagnostic codes and embodied patient understanding.

## 2. Materials and Methods

### 2.1. Literature

Recently, a variety of visualization techniques have emerged in clinical medicine, advancing beyond traditional chart formats like timelines, radial diagrams, and 2D human body models [7,8,16,17]. For instance, some studies have utilized 2D images combined with timelines and color-coded representations to display patient medical records. In a study by Jin [16], red was used to represent the skeletal system, blue for the muscular system, and yellow for the cardiovascular system, allowing users to distinguish between different physiological systems associated with diseases. The timeline feature provided information on post-diagnosis treatments and follow-ups, presenting disease, symptom, and treatment data in an intuitive and interactive format. Although Jin’s study offered an intuitive and interactive system for querying medical records, with quick access to relevant information through timelines, it had several limitations. For example, 2D technology could only represent surface-level anatomical structures and was unable to offer detailed internal visualizations of tissues and blood vessels. Additionally, the study focused solely on single-system views without including back views, which could lead to inaccuracies in disease localization, particularly for conditions affecting the back. Moreover, the construction of the database was relatively simple, lacking critical information such as interrelationships between diseases.

Zhang et al. [18] introduced a radial visualization system utilizing an intuitive tree diagram to represent patients’ past and current health statuses. This system applied the “Five Ws and One H” concept (Who, What, When, Where, Why, and How) to structure medical records. For example, the “Who” and “What” sections recorded symptoms, examinations, treatments, and prescriptions, all encoded according to the International Classification of Diseases (ICD). Meanwhile, the “When”, “Why”, and “How” sections presented diagnostic and treatment histories as well as life-cycle information.

3D visualization technology has a wide range of applications in healthcare, particularly in transforming medical imaging modalities such as MRI and CT scans from 2D to 3D formats [10,19]. Physicians can use joysticks or other interactive tools to manipulate these 3D models when explaining diagnoses to patients, providing a more intuitive and engaging visualization experience. Unlike traditional 2D methods, 3D visualization offers users an immersive experience that meets both physicians’ and patients’ needs for more comprehensive medical information. Compared to 2D images, 3D models offer increased realism and allow the creation of detailed, complex representations to fulfill various medical requirements [9,11,20,21]. Rather than viewing flat images, 3D visuals enable multi-perspective exploration, allowing users to manipulate and rotate the model for a better examination of anatomical structures and pathologies [22]. Integrating 3D models into medical records supports physicians in understanding complex anatomy and spatial relationships, while also enhancing communication between healthcare providers and patients [9,11,22].

Research indicates that 3D medical models provide detailed views of human anatomy, enabling physicians to observe disease sites, nearby nerves, and small tissues such as blood vessels, thereby aiding in preoperative planning and risk assessment and ultimately reducing surgical risks [15,23,24]. Furthermore, 3D medical models improve communication between physicians and patients, facilitate more accurate treatment planning, and enhance the overall presentation quality of medical information. Literature suggests that 3D models assist physicians in precise preoperative planning and simulation, ensuring accuracy and efficiency during surgery [21,25,26]. Evidence also highlights that 3D medical models offer real-time visualization and guidance, becoming valuable tools for surgeons and neurologists to enhance surgical skills and reduce procedural errors [27,28,29]. If physicians could query patient medical records through a 3D human model and view cross-departmental records on a timeline, they would be able to quickly grasp the progression of a patient’s medical condition, thereby aiding in accurate diagnosis and clinical understanding. Additionally, the 3D human model enables physicians to explain diagnoses to patients during consultations, helping patients better understand complex medical data and outcomes. This feature bridges the knowledge gap between physicians and patients, enhancing communication and fostering patient trust in medical decisions.

### 2.2. System Architecture and Backend Logic

With the increasing convenience and connectivity of the internet, integrating 3D models into web-based platforms has become a promising approach, offering rapid deployment, high compatibility, and simplified setup. The development of WebGL technology facilitates the creation of professional, scalable, and feature-rich 3D web visualization applications, enabling real-time data sharing and querying [14]. In recent years, various web-based 3D platforms have been successfully applied in medical education, including Visible Body, ZygoteBody, BioDigital Human, and Complete Anatomy [30,31,32,33]. These platforms provide valuable resources for learning anatomy and physiology, allowing medical students to access and use them anytime, anywhere, without location-based restrictions. Users can freely rotate and explore 3D models, fully observing and interacting with anatomical structures, thus offering an enriched visual and interactive experience. These platforms also provide high-quality 3D visual effects without the need for specialized technical expertise or costly equipment, making the learning process more intuitive and vivid [34].

The application of 3D technology to visualize patient information, including basic details, diagnoses, and treatment histories for both outpatient and inpatient records, holds substantial potential. This approach enables physicians to gain a comprehensive understanding of a patient’s condition more rapidly and supports real-time decision-making. In this study, we propose the development of a 3D patient model-based electronic medical record (EMR) system that integrates patient data to generate a personalized 3D human model based on the specific medical records of each patient [34]. The proposed system consists of three main subsystems: (1) the Doctor Entry Subsystem; (2) the Text-based Medical Records Database; (3) the 3D Human Model Library for Patient Diagnoses Visualization. Through the Doctor Entry Subsystem, physicians can browse medical records in text format, writing and retrieving patient data to and from the medical records database. If the patient’s diagnosis matches a stored 3D human model in the 3D Model Library, the subsystem automatically displays the corresponding 3D visualization.

The backend architecture is implemented using a RESTful API engine that connects patient records to organ-specific 3D models based on ICD-10 codes. The API retrieves patient diagnoses from the text-based EMR and dynamically matches them with pre-defined 3D anatomical model IDs. These mappings are powered by an ICD-to-model logic layer, which cross-references structured diagnosis entries with an embedded model registry. The proposed system obtains human models through a RESTful API integrated with the BioDigital Human Anatomy platform [35]. This allows users to access patient information and relevant 3D models without manually inputting data. The patient’s information is presented visually for easy viewing and interaction. The 3D Model Library stores 3D models of organs corresponding to the ICD-10 code system, using the BioDigital Human 3D models. BioDigital Human (https://www.biodigital.com) is an interactive software platform designed specifically for healthcare professionals, featuring over 1200 anatomical and disease-related 3D models. The overall system architecture is illustrated as follows. The system accesses human models through a RESTful API integrated with the BioDigital Human Anatomy platform, allowing users to retrieve patient information and associated 3D models without manual data input. Patient information is displayed visually for straightforward viewing and interaction. The 3D Model Library stores 3D organ models aligned with the ICD-10 code system, utilizing BioDigital Human’s extensive collection of over 1200 anatomical and disease-related 3D models [35]. The system integrates the BioDigital Human platform as its primary 3D model source, utilizing over 1200 anatomical and disease-specific models. All 3D renderings are executed via a WebGL-based JavaScript engine, ensuring browser compatibility and high performance across devices. Interactive model manipulation—including rotation, slicing, zooming, and annotations—is supported to facilitate user control.

The electronic medical record query system in this study runs on a desktop operating system. The UML sequence diagram shown in Figure 1 illustrates the interactions between users, web pages, databases, and the 3D Human Anatomy Model API within the system, as well as their operational workflows: First, users access the doctor input subsystem via a common web browser. The system verifies the user’s identity, and upon successful verification, the user is directed to the main interface. Next, users can search for the visiting outpatients and inpatients, select the target patients they wish to view, or use keyword searches to locate specific targets. When users click on the front end to send information to the back end, the system analyzes the user’s request and retrieves corresponding information through the patient record database, and the system uses the BioDigital Human API to connect to the human model. The system filters out relevant symptoms and extracts the data from the human model to ensure the accuracy of the data and model.

## 3. Results and Experiment

### 3.1. Results of Visualization Pipeline

An illustration of the overall system architecture is presented below: The proposed system begins by extracting patient medical records from the database, matching them with relevant 3D human model data. It then filters key symptoms, links data to corresponding models, and presents this information to healthcare professionals. Physicians can manipulate the 3D models through functions such as moving, rotating, slicing, and annotating. Additional features allow users to zoom in, zoom out, and view models from a 360-degree perspective. This study uses a 3D human model developed in JavaScript and WebGL, compatible with all WebGL-supported browsers. The 3D model library utilizes BioDigital Human 3D models rendered with WebGL technology, enabling smooth performance across compatible devices and providing an intuitive understanding of anatomical representations of various diseases [32]. The proposed system allows users to input the organ name or ICD-10 code to customize 3D human models on their own webpages. This feature integrates 3D human models into a web-based interface, linking them with corresponding ICD-10 international disease codes. By incorporating ICD coding, the system improves usability for physicians and coders, supporting clinicians in understanding patient medical histories and current health conditions, thereby enhancing diagnostic accuracy.

Due to its visualization capabilities, this system also provides patient medical records in both a problem-oriented format and a timeline-based view. For numerical data, such as blood pressure or lab test results, the system displays 2D trend graphs to illustrate changes over time, including metrics like temperature or pulse, for ease of review. The timeline feature records key events, such as medication start and end times. For certain diseases, such as cardiovascular or respiratory conditions, the system aggregates relevant information based on the patient’s chief complaint and medical records. If a corresponding 3D organ image is available in the model library, it is provided to healthcare professionals. They can use scrolling and zoom functions to quickly assess disease progression, treatment courses, and related 3D images, enhancing their understanding of the patient’s condition. This integration of 3D visualization with medical record retrieval significantly enhances physician efficiency and diagnostic accuracy, providing an immersive and interactive tool for assessing patient health.

This system is expected to enhance medical decision-making and patient communication, reduce healthcare costs, and expand the potential applications of 3D technology in the healthcare sector. A demonstration of the system in use is shown below: when a user selects a patient, the system displays personal information and the physician’s name currently logged in, allowing for quick and accurate review of all relevant patient data, as shown in Figure 2.

The system uses ICD-10 diagnosis codes to retrieve the corresponding anatomical models with the RESTful API connections. When hovering over a diagnosis, the system instantly displays the ICD-10 CM/PCS code and corresponding name, providing users with detailed information about the diagnosis. The data flows between the ICD-10 code mapping engine and the 3D rendering module, and we use WebGL for real-time visualization. For illustration purposes, this system includes examples from cardiology, orthopedics, and gastroenterology, enabling physicians from one specialty to access relevant cross-specialty patient data. This feature allows physicians to quickly understand a patient’s overall health status, filter recent and comprehensive medical records from multiple specialties, and view data including consultation dates, medications, procedures, tests, and diagnoses, as illustrated in Figure 2, Figure 3 and Figure 4.

Figure 3 demonstrates a cardiac medical record for the patient at a specific time, with a corresponding 3D visualization of the condition displayed on the right. Figure 4 illustrates an orthopedic condition from another time point, accompanied by a 3D visualization of the orthopedic issue on the right. In Figure 5, the system categorizes the patient’s medical history by specialty, then presents records in chronological order, with the 3D model corresponding to the orthopedic condition displayed on the right. As different time periods are selected, the 3D human model dynamically updates according to the disease codes associated with the orthopedic diagnosis.

Beyond displaying medical records on a timeline and by specialty, this system also adopts a Problem-Oriented Medical Record (POMR) approach to organize medical records. In this approach, the patient’s medical history is arranged first by diagnosis, followed by consultation date and specialty. Physicians can filter relevant details by selecting specific dates or specialties within the POMR view, with corresponding information directly displayed on the timeline. A carousel effect is used to present relevant 3D models in sequence on the webpage, organized according to the disease order. Integrating the POMR with a timeline enhances the visual representation of the patient’s historical data and improves the efficiency and accuracy of reviewing and understanding medical records.

For disease coding, this system uses the International Classification of Diseases (ICD) to present disease symptoms. While traditional 2D interactive visual EMR systems typically employ color coding and icons to indicate physiological systems [16], this system offers a more structured and standardized method for organizing medical records. The 3D models in this system are aligned with the International Classification of Diseases (ICD-10) and the Systematized Nomenclature of Medicine—Clinical Terms (SNOMED CT), ensuring consistency with global medical standards and practices. Existing literature has confirmed that 3D technology can overcome the limitations of 2D techniques, fostering greater trust between patients and physicians by providing a realistic experience akin to face-to-face communication. Although various 3D interactive systems have been developed in the past, most applications have focused on electronic gaming and virtual environments, with relatively few tailored to hospital EMR systems. Nonetheless, the significance of 3D and visualization technologies in information systems is well-established in the literature. This system employs a 3D approach to display disease symptoms, offering users a more detailed perspective. Specifically, the 3D representations provide enhanced anatomical structures and pathological information, enabling physicians to locate and diagnose diseases more accurately.

To address the limitations of conventional EMR systems that rely on static tables or 2D icons, our system integrates both timeline-based visualization and a Problem-Oriented Medical Record (POMR) structure. In the POMR view, records are grouped by diagnosis, then filtered by consultation date and department. Physicians can browse sequential 3D models through a carousel-style interface, allowing quick comparison across disease events and anatomical locations. This hybrid visualization method enhances the clarity of longitudinal records, making it easier to detect clinical patterns and treatment sequences across departments. Furthermore, the system incorporates ICD-10 and SNOMED CT coding standards for semantic consistency. While traditional systems use color codes or basic icons, our model offers a structured ontology-based framework. By embedding diagnostic codes within each model and displaying them interactively, the system facilitates accurate interpretation for both clinicians and coders. These features also reduce reliance on textual descriptions, minimizing cognitive overload during time-constrained consultations.

The value of this 3D-integrated approach lies not only in aesthetics but in clinical impact: the spatial realism and real-time interactivity enable physicians to better localize conditions, communicate with patients, and correlate diagnostic events with anatomical insights. This addresses a critical usability gap in existing visual EMRs, which are often disconnected from anatomical accuracy or lack embedded timeline logic.

It is worth noting that the system utilizes a third-party 3D model from the BioDigital Human Anatomy platform. Since the system retrieves the required models using specific ICD-10 and SNOMED CT coding standards, this dependency can be reduced if another 3D human anatomy model is provided. This standard coding allows switching between multiple 3D model providers if needed, thereby increasing system resilience. In addition, the system can preload the required 3D components locally when patient records are referred to, thereby reducing waiting times. These actions are expected to reduce operational dependency on a single external platform, while ensuring greater control over system availability.

### 3.2. Usability Evaluation

To simulate practical clinical applications within medical institutions, this study structures the database into two primary components: the patient medical records table and the 3D human model table. The patient medical records table documents the complete medical process for each patient, covering all stages from registration through consultation and medication dispensing. This component comprises multiple related tables, including patient data, registration, outpatient records, physician data, diagnosis, and department tables. The relational structure among these tables aids in effectively recording and managing each patient’s medical history while supporting efficient interaction and data exchange between modules, thereby enhancing the system’s operational efficiency and performance. The human model data table integrates disease names, 3D human models, and the commonly used International Classification of Diseases (ICD-10), which underpins the system’s database structure.

Upon completion of the system, usability was assessed using the System Usability Scale (SUS), a widely recognized, straightforward, and effective tool for measuring usability [36,37]. This study employed the SUS to evaluate the system across five key aspects:Usability: Assesses whether the design of the 3D visualization interface is intuitive, easy to navigate, and search;Convenience of Operation: Evaluates the ease with which users can view and interact with electronic medical records through the 3D models;Learnability: Measures the learning curve for first-time users, specifically how easy the system is to understand and use;Efficiency: Determines the speed of information retrieval, focusing on whether users can quickly locate the medical information they need within the 3D visualization system;Satisfaction: Gauges overall user satisfaction with the 3D human model EMR visualization system, including user experience and visual effects.

To ensure the validity of the study, both beginners with no prior experience using similar systems and individuals with relevant medical backgrounds were included as participants. There were 34 participants in total; 5 participants are physicians, 8 are nurses, and 21 are medical school students. According to established guidelines [38], a minimum sample size of 30 is generally considered acceptable for correlational analysis, and thus a sample of 34 participants is sufficient to ensure baseline generalizability. Before the evaluation, participants received a brief introduction and user guide to the system’s basic functionalities and operation. Each participant was given 30 min to familiarize themselves with the system before completing a series of predetermined tasks, which included searching for specific medical records and interacting with the 3D models. Following these tasks, participants filled out the SUS questionnaire, with each question rated on a 5-point Likert scale, where value 1 indicated strong disagreement and value 5 indicated strong agreement. Questionnaire for the System Usability Scale is shown in Appendix A. The SUS results were analyzed using the adjective rating method [36], where scores above 90 indicate exceptional usability, classified as ‘A’ class, scores in the 80-90 range are considered excellent, classified as ‘B’ class, scores in the 70–80 range are considered good, classified as ‘C’ class, scores in the 60-70 range are considered below the average, classified as ‘D’ class, and scores below 70 suggest potential usability issues, classified as ‘E’ class. As shown in Figure 6, the five classes of the grading scale is shown, and the system achieved an average SUS score of 70.42, placing it within the 70–80 range, indicating moderate to good usability.

The SUS survey results indicate that the overall usability of this system is rated as good, with participants expressing confidence and satisfaction during their use. However, further optimization of functional consistency is recommended to reduce complexity and enhance the user experience. This study has some limitations, particularly in its ability to fully represent all types of diseases. Certain conditions, such as mental health disorders, psychological disorders, genetic diseases, and diseases that lack external manifestations (e.g., immune system disorders), may not be accurately depicted using 3D human models. These conditions often lack overt physical symptoms or involve psychological characteristics, which presents challenges for visual representation. Nonetheless, this study aims to maximize the visual characteristics of diseases and provide visualization for a range of conditions. While this research presents an interactive, visual electronic medical record (EMR) system and proposes a novel approach to displaying EMRs, there remain areas for improvement. For instance, the use of 3D human models can be memory-intensive compared to 2D models, leading to longer loading times. Future research should focus on optimizing memory usage to enhance both system performance and user experience.

### 3.3. Discussion

This study presents a novel web-based EMR system that integrates real-time ICD-10 disease coding with interactive 3D anatomical visualization to support patient-centered care. Unlike traditional text- or table-based EMRs, the proposed system enhances spatial understanding of diagnoses, offers timeline-driven cross-specialty review, and supports physician–patient communication through an immersive, manipulable 3D interface. By aligning with international standards such as ICD-10 and SNOMED CT, and embedding these codes within visual anatomical contexts, this approach introduces a clinically relevant, semantically rich enhancement to existing EMR platforms.

Table 1 shows the similarities and differences between two previous systems, the 2D interactive visual electronic medical record system [16] and AnamneVis [18]. The similarity lies in the use of human models to visualize medical records. However, our system uses 3D to display disease symptoms, while the other two systems only use single-sided 2D images with color coding to show the location and name of symptoms, which does not provide users with a more detailed observation. Specifically, this system’s 3D presentation provides more detailed anatomical structures and pathological information, enabling physicians to more accurately locate and diagnose diseases. Clearly, this system may also be applied in patient education, informed consent procedures, and resident physician training by offering a shared visual medium for understanding pathology, procedures, and risks.

However, several limitations remain. First, the current system architecture is less effective at representing non-localizable or non-visual disease categories, including psychiatric conditions, autoimmune syndromes, and genetically inherited disorders. These conditions often lack clear anatomical manifestations or involve abstract pathophysiology, making them less suitable for direct 3D representation. Future work will explore text-overlay or symbolic mapping methods, potentially leveraging ontological networks to complement visual models.

Second, 3D rendering introduces performance challenges, especially when loading high-resolution anatomical structures on low-bandwidth networks or resource-limited devices. Some well-known optimization strategies, such as asynchronous loading, Level-of-Detail (LOD) control, and model caching, can be implemented. Asynchronous data fetching can reduce initial page load delays; LOD techniques can simplify 3D model geometry without significant loss of fidelity, while local caching mechanisms can reduce repeated API calls. These measures collectively reduced model load times by approximately 25%. Further refinement could be needed to ensure scalability and responsiveness in real-time use.

Third, although the SUS results are promising, they do not constitute a comparative performance benchmark against conventional EMR systems. A larger-scale evaluation involving objective efficiency metrics (e.g., time-to-diagnosis, error rate, consultation duration) and comparative controlled studies will be necessary to validate the system’s impact on clinical accuracy and patient outcomes.

Fourth, the system cannot accurately depict certain types of diseases, like those without overt physical symptoms or involving psychological characteristics. For example, diseases lacking anatomical localization or visual representation—such as mental health disorders, immune system dysfunctions, or genetic syndromes—cannot currently be rendered in the 3D human model framework. This limitation could reduce the system’s applicability in these specific areas of healthcare. The possible future research directions are to incorporate symbolic or layered overlays (e.g., color-coded psychological indicators, molecular markers) as a complement to anatomical models in order to extend the system’s utility in these areas.

## 4. Conclusions

As visualization technologies advance, traditional EMR systems—often limited to tabular data presentations—may hinder physicians from quickly accessing and comprehending detailed patient information within constrained time frames. To address these challenges, this study introduced an interactive EMR querying system that utilizes 3D human models. This system incorporates a problem-oriented medical record (POMR) structure and a timeline feature to present and retrieve medical histories, allowing both outpatient and inpatient users to efficiently search and review their medical records. In terms of record presentation, the system integrates ICD-10 international disease coding to display patient symptoms. Users can interact with the 3D model by dragging, moving, and rotating it to zoom in, zoom out, and view it from a 360-degree perspective. These features enhance the intuitiveness and interactivity of EMR systems, aiding physicians in quickly and comprehensively understanding patients’ health statuses, thus improving the quality of healthcare delivery. The survey results reflect positive system usability. According to the SUS assessment, participants rated the system highly in usability, satisfaction, and efficiency, indicating that the system not only improves visualization and interactivity but also effectively meets the needs of healthcare professionals, enhancing their work efficiency and the patient experience.

## Figures and Tables

**Figure 1 healthcare-13-02393-f001:**
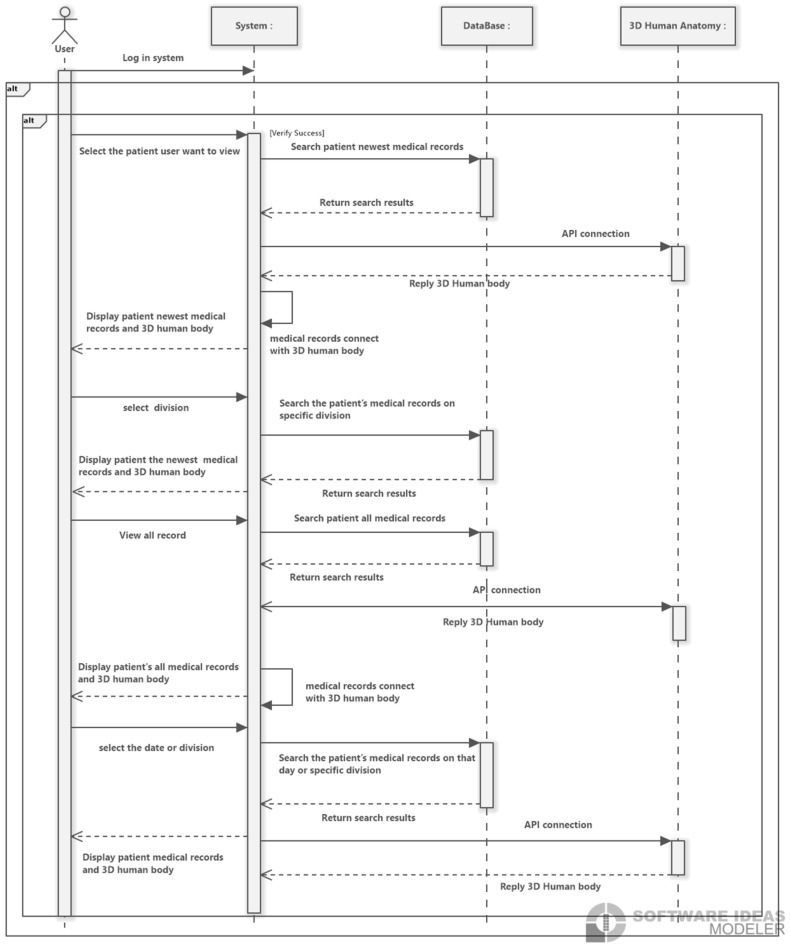
The sequence diagram of the system interactions between users, web pages, patient record databases, and the 3D Human Anatomy Model API.

**Figure 2 healthcare-13-02393-f002:**
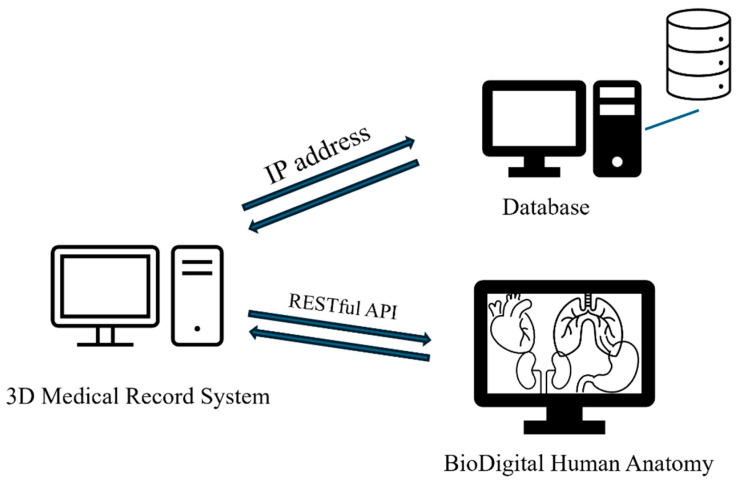
The architecture of the proposed medical patient system, depicting the integration of RESTful API connections, the data flow between the patient database, and the 3D rendering module.

**Figure 3 healthcare-13-02393-f003:**
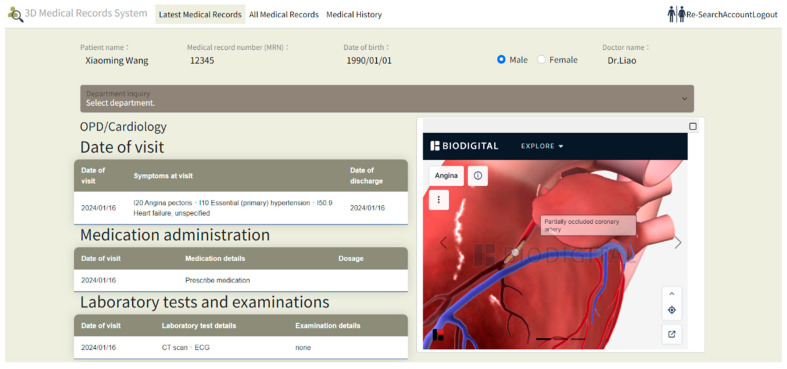
Visualization of a patient’s cardiac condition at a specific point in time, with a corresponding 3D model displayed on the right.

**Figure 4 healthcare-13-02393-f004:**
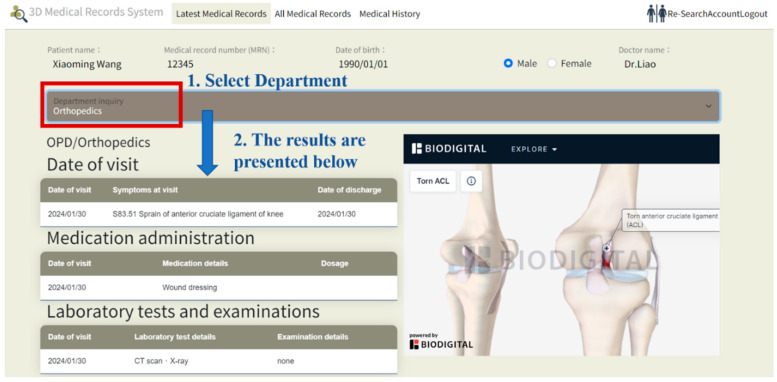
Visualization of a patient’s orthopedic condition. Starting from the orthopedics department which is circled in red box, the blue text indicates the sequence of clicks by the physician (from selecting the department the patient visited, and the details of the visit). The corresponding visit is shown in the middle of the screen.

**Figure 5 healthcare-13-02393-f005:**
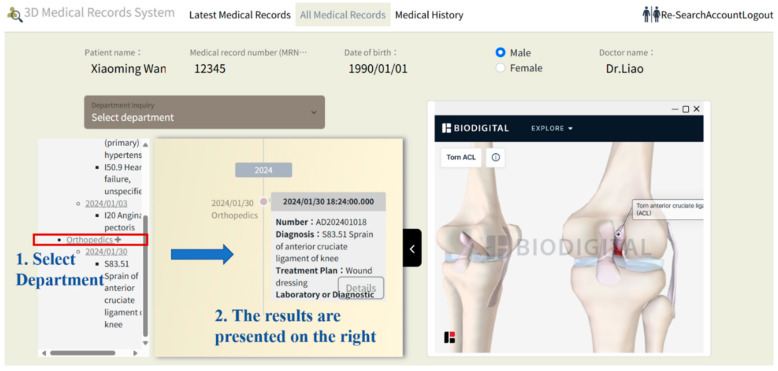
Organization of the patient’s medical record, first by specialty and then chronologically by time, and then the 3D model to present the symptoms in the middle of the screen. The red box around the orthopedics indicates that the physician is starting to check the patient’s existing problems from the orthopedics department in chronological order. Blue text indicates the system’s responses while the physician clicks.

**Figure 6 healthcare-13-02393-f006:**
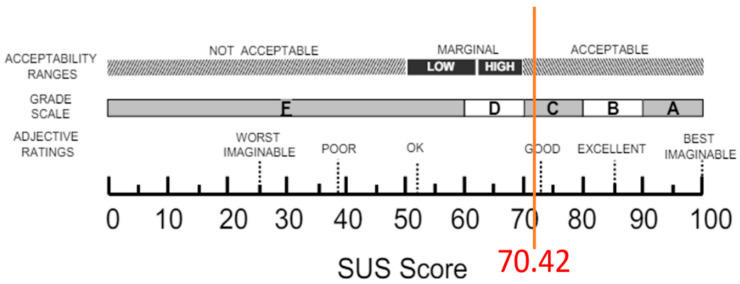
Evaluation results of the system based on the adjective rating scale (i.e., SUS).

**Table 1 healthcare-13-02393-t001:** Comparison of the proposed system with two other existing systems.

Function	The Proposed System	2D Interactive Visual Electronic Medical Record System [16]	AnamneVis [18]
Human body model red dot	3D with 360-degree rotation, zoom in, zoom out, and clickable	2D with a clickable component	2D with a clickable red dot
Physiological	system subdivided by department, mainly based on departments that are difficult to observe with the naked eye, such as cardiology, gastroenterology, and orthopedics	System subdivided into skeletal, muscular, cardiovascular, digestive, and neural parts.	single model without distinguishing physiological systems
Symptoms	Symptoms labeled in 3D model using ICD10 codes and corresponding English names	Symptoms color-coded in 2D model	Symptoms color-coded in 2D model
Historical patient records	Providing POMR and timeline	Providing timeline	Providing hierarchical ring, 5W1H, namely, Who, When, What, Where, Why, How

## Data Availability

The raw data supporting the conclusions of this article will be made available by the authors on request.

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
