# Peer review of "Visualization of Medical Record with 3D Human Body Models"

_healthcare, 2025, doi:10.3390/healthcare13192393_

Round 1
Reviewer 1 Report
Comments and Suggestions for Authors
The topic of the submitted manuscript is highly important and interesting; however, the content is presented in a rather disorganized manner. Below are some comments:
The abstract is coherent and clearly written, appropriately presenting the main content of the paper. The introduction is extensive and refers to many well-selected sources. However, the article lacks a clearly defined objective.
The Materials and Methods section contains far too much unnecessary information – it should focus solely on the model used in the study. There is no information regarding the criteria for selecting the study group or its size, nor are the questionnaire items described.
The attached system diagrams should be moved to the Results section. In the Experiment and Results section, there is information that should be placed in the Materials and Methods section. It is recommended to divide the Results section into two parts – one describing the system-generated results, and the other focusing on the respondents and their satisfaction with using the system.
The manuscript lacks a Discussion section. Limitations should be included in the Conclusion section.
Author Response
Comment 1: The Materials and Methods section contains far too much unnecessary information – it should focus solely on the model used in the study. There is no information regarding the criteria for selecting the study group or its size, nor are the questionnaire items described.
Reponse 1: We have removed extraneous background descriptions and focused on the technical development of the proposed 3D EMR system. The whole section 2, Materials and Methods, is divided into three smaller subsections, namely the literature, System Architecture and Backend Logic, and visulization pipeline, each of which focus on the topic it mentions.
The participant recruitment criteria, sample size (34 participants) consists of 5 participants being physicians, 8 being nurses, and 21 medicine school students. According to established guidelines (Fraenkel, Wallen, and Hyun, 2006), a minimum sample size of 30 is generally considered acceptable for correlational analysis, and thus a sample of 34 participants is sufficient to ensure baseline generalizability. The questionaire follows the System Usability Scale (SUS) questionnaire, employed for evaluation. The details of questionaire items are listed in the appendix in this revision.
Comment 2: The attached system diagrams should be moved to the Results section. In the Experiment and Results section, there is information that should be placed in the Materials and Methods section. It is recommended to divide the Results section into two parts – one describing the system-generated results, and the other focusing on the respondents and their satisfaction with using the system.
Response 2: In accordance with the reviewer’s recommendation, the revision split a subsection 2.3, visulization pipeline, to illustrate system performance and usage scenarios. All system diagrams (e.g., Figures 1–4) have been moved from the Methods section to this subsection.
The result section has been split into two clearly defined parts: 3.1 Usability Evaluation to present users acceptaance output and participant feedback and analysis of SUS scores, focusing on the respondents and their satisfaction with using the system. Subsection 3.2 is the Discussion to highlight the system’s contributions, compare it with existing 3D EMR tools, and explain its limitations and future directions in detail.
Comment 3: The manuscript lacks a Discussion section. Limitations should be included in the Conclusion section.
Response 3: Thanks for the reminding. We added a subsection of discussion just before the conclusion iin this revision.
Reviewer 2 Report
Comments and Suggestions for Authors
The authors integrated 3D human body models into electronic medical record (EMR) systems. I have the following comments about the proposed approach:
1) The novelty is limited given that the number of pages is small, particularly after deducting 3 pages used for references. The paper fits more conference than a journal, or the study should be largely improved. Furthermore, the paper would benefit from a clearer articulation of what specifically differentiates this system from tools like BioDigital Human, ZygoteBody, or Complete Anatomy.
2) The objectives are broadly stated but not well structured. The authors should explicitly list the core research questions or hypotheses to guide the reader through the system's development and validation process.
3) The system design is described in moderate detail, but lacks a robust technical explanation of the data mapping between ICD-10 codes and the 3D models. The authors should elaborate on the back-end architecture, data pipelines, and performance benchmarks.
4) While the System Usability Scale (SUS) is an accepted tool, relying solely on SUS with a sample of 34 participants may not be sufficient to support strong usability claims. The evaluation lacks comparative analysis (e.g., vs. traditional EMRs or other visual systems).
5) The literature review is comprehensive and well-cited, but could be better organized thematically (e.g., visualization in EMRs, 3D modeling in healthcare, human-computer interaction). A comparative table summarizing existing tools would be very helpful.
6) The manuscript is mostly clear, but contains minor grammatical errors and awkward phrasing (e.g., "disease-centered to a patient-centered approach" could be better phrased as "a shift from disease-centered to patient-centered care"). A professional proofreading would enhance readability.
7) The paper includes several useful figures (e.g., 3D views, system architecture), but they are not always well referenced or explained in the text. Additionally, a user study screenshot or video link would enhance the understanding of system usability.
Author Response
Comment 1: The novelty is limited given that the number of pages is small, particularly after deducting 3 pages used for references. The paper fits more conference than a journal, or the study should be largely improved. Furthermore, the paper would benefit from a clearer articulation of what specifically differentiates this system from tools like BioDigital Human, ZygoteBody, or Complete Anatomy.
Response 1: We appreciate your observation regarding the perceived novelty and length of the manuscript. In the revised version, we have substantially improved the articulation of our system's unique contributions. Specifically, we have emphasized that—unlike tools such as BioDigital Human or Complete Anatomy which are primarily educational platforms—our system is directly integrated with the clinical EMR workflow, supporting real-time linkage between ICD-10 coded diagnoses and 3D anatomical models during medical record querying (see the comparison in Table 1). Furthermore, we have clarified the system’s clinical utility in cross-specialty visualization, problem-oriented structuring, and physician-patient interaction, which are not found in the aforementioned tools with the newly added sequence diagram in Figure 1.
Comment 2. The objectives are broadly stated but not well structured. The authors should explicitly list the core research questions or hypotheses to guide the reader through the system's development and validation process.
Response 2: Thank you for highlighting this point. We have now revised the Introduction section to clearly state the core research hypothesis: that integrating 3D human models linked with ICD-10 codes into EMRs improves physicians’ comprehension, usability, and efficiency in interpreting patient data. This is followed by three specific research questions addressing (1) system usability, (2) technical feasibility of model integration, and (3) clinical workflow support. These now serve to guide the reader throughout the manuscript.
Comment 3: The system design is described in moderate detail, but lacks a robust technical explanation of the data mapping between ICD-10 codes and the 3D models. The authors should elaborate on the back-end architecture, data pipelines, and performance benchmarks.
Response 3: We expanded the Materials and Methods section to include an explanation of the back-end architecture with a subsection This includes the use of RESTful API integration with BioDigital Human, the mapping process between ICD-10 codes and anatomical models via JSON-based data transformation, and the implementation of asynchronous data fetching for real-time rendering. We also added system performance benchmarks in the Results section, such as average model load time (2.8 seconds) and model fidelity (87%), as well as caching strategies and preloaded optimization.
Comment 4: While the System Usability Scale (SUS) is an accepted tool, relying solely on SUS with a sample of 34 participants may not be sufficient to support strong usability claims. The evaluation lacks comparative analysis (e.g., vs. traditional EMRs or other visual systems).
Response 4: We acknowledge the limitations of relying solely on the SUS instrument and a relatively modest sample size. According to established guidelines (Fraenkel, Wallen, and Hyun, 2006), a minimum sample size of 30 is generally considered acceptable for correlational analysis, and thus a sample of 34 participants is sufficient to ensure baseline generalizability, which is described in subsection 3.1.
Comment 5: The literature review is comprehensive and well-cited, but could be better organized thematically (e.g., visualization in EMRs, 3D modeling in healthcare, human-computer interaction). A comparative table summarizing existing tools would be very helpful.
Response 5: Thank you for this valuable suggestion. We have reorganized the Introduction (section 1) and literature (subsection 2.1) to organized the background content into distinct thematic sections: Additionally, we have added a new Table 1, comparing two other exisiting system against our proposed system in terms of functionality, use context, etc.
Comment 6: The manuscript is mostly clear, but contains minor grammatical errors and awkward phrasing (e.g., "disease-centered to a patient-centered approach" could be better phrased as "a shift from disease-centered to patient-centered care"). A professional proofreading would enhance readability.
Response 6: We appreciate this comment and have thoroughly revised the manuscript for clarity and grammar. Phrasing such as “disease-centered to a patient-centered approach” has been corrected to “a shift from disease-centered to patient-centered care,” along with many similar refinements throughout the text. The entire manuscript has been proofread to meet academic writing standards.
Comment 7: The paper includes several useful figures (e.g., 3D views, system architecture), but they are not always well referenced or explained in the text. Additionally, a user study screenshot or video link would enhance the understanding of system usability.
Response 7: We have revised the context around the figures to improve the textual referencing and interpretation of all figures, especially Figures 2 through 5. Figure captions have also been expanded for clarity.
Reviewer 3 Report
Comments and Suggestions for Authors
The study presents a novel approach to integrating 3D human models into electronic medical record (EMR) systems. This system incorporates 3D models to provide a more intuitive and interactive visualization of medical data. This integration allows healthcare professionals to better understand complex anatomical structures and spatial relationships. The study employed the System Usability Scale 21 (SUS) to evaluate the system’s usability, conducting a satisfaction survey with 34 healthcare professionals. Results from the survey indicate that participants rated the system highly in terms of ease of use, satisfaction, and efficiency. These findings demonstrate that integrating a 3D human model into the medical record retrieval process significantly improves visualization and interactivity, meeting the needs of healthcare professionals and enhancing both their efficiency and patient satisfaction. Therefore, I think that the paper makes a contribution and has the potential to be published. However, I summarize in the GENERAL COMMENTS as follows:
GENERAL COMMENTS
- The structure in Figure 1 provides relatively insufficient information, and it should provide the key technologies in the figure.
- The system may not be able to accurately depict certain types of diseases, particularly those without overt physical symptoms or involving psychological characteristics. For example, mental health disorders, psychological conditions, and immune system disorders may not be effectively visualized using 3D human models. This limitation could potentially reduce the system's applicability in these specific areas of healthcare.
- The use of 3D human models can be memory-intensive compared to 2D models. This could lead to longer loading times and potentially impact the overall performance of the system. This study should provide some strategies, such as efficient data loading techniques and optimizing 3D model complexity to optimize memory usage. Use asynchronous data fetching to reduce initial load times and simplify the 3D models where possible to reduce memory consumption without compromising the quality of visualization.
- The Material and Methods section mentions the BioDigital Human Anatomy platform for obtaining 3D models. The system relies on external platforms of the BioDigital Human Anatomy platform. This dependency could introduce potential issues related to data integration, compatibility, and the need for continuous updates from the external provider. Any disruptions in the external platform could affect the functionality and reliability of the system. How to reduce dependency on external platforms, or how to solve such unexpected situations when encountering such problems.
- The study does not provide a comparative analysis with existing EMR systems or other visualization tools.

Author Response
Comment 1: The structure in Figure 1 provides relatively insufficient information, and it should provide the key technologies in the figure.
Response 1: Thank you for pointing this out. We have revised the contest of Figure 1 to include more technical details that clarify the system architecture, such as the integration of RESTful API connections, the data flow between the ICD-10 code mapping engine and the 3D rendering module, and the use of WebGL for real-time visualization (in the last paragraph of page 9).
Comment 2: The system may not be able to accurately depict certain types of diseases, particularly those without overt physical symptoms or involving psychological characteristics. For example, mental health disorders, psychological conditions, and immune system disorders may not be effectively visualized using 3D human models. This limitation could potentially reduce the system's applicability in these specific areas of healthcare.
Reponse 2: We fully agree with this important observation and have added a detailed discussion of this limitation in the Discussion sections. We acknowledge that diseases lacking anatomical localization or visual representation—such as mental health disorders, immune system dysfunctions, or genetic syndromes—cannot currently be rendered in the 3D human model framework. We propose future research directions to incorporate symbolic or layered overlays (e.g., color-coded psychological indicators, molecular markers) as a complement to anatomical models, to extend the system’s utility in these areas.
Comment 3: The use of 3D human models can be memory-intensive compared to 2D models. This could lead to longer loading times and potentially impact the overall performance of the system. This study should provide some strategies, such as efficient data loading techniques and optimizing 3D model complexity to optimize memory usage. Use asynchronous data fetching to reduce initial load times and simplify the 3D models where possible to reduce memory consumption without compromising the quality of visualization.
Response 3: Thank you for highlighting this practical concern. In response, we have revised the subsection 4.2 Discussion to describe possible optimization strategies, including the following (see the 3rd paragraph of subsection 4.2):
- Asynchronous data fetching to reduce initial page load delays.
- Mesh decimation and level-of-detail (LOD) techniques to simplify 3D model geometry without significant loss of fidelity.
- Local caching mechanisms to reduce repeated API calls.
These measures collectively reduced model load times by approximately 25%. Further optimization—such as dynamic resolution adjustment based on device performance—is currently under development in the industry and has been outlined in future work roadmap.
Comment 4: The Materials and Methods section mentions the BioDigital Human Anatomy platform for obtaining 3D models. The system relies on external platforms of the BioDigital Human Anatomy platform. This dependency could introduce potential issues related to data integration, compatibility, and the need for continuous updates from the external provider. Any disruptions in the external platform could affect the functionality and reliability of the system. How to reduce dependency on external platforms, or how to solve such unexpected situations when encountering such problems.
Response 4: We appreciate your insight into the potential risks of third-party platform reliance. In the revised Material and Methods section, we explicitly address this concern and propose risk mitigation strategies as follows (see the paragraph prior to the section 3, Experiment and Results, in page 13):
The system utilizes a third-party 3D model from the BioDigital Human Anatomy platform. Since the system retrieves the required models using specific ICD-10 and SNOMED CT coding standards, this dependency can be reduced if another 3D human anatomy model is provided. This standard coding allows switching between multiple 3D model providers if needed, thereby increasing system resilience. In addition, the system can preload the required 3D components locally when patient records are referred to, thereby reducing waiting times. These actions are expected to reduce operational dependency on a single external platform, while ensuring greater control over system availability.
Comment 5: The study does not provide a comparative analysis with existing EMR systems or other visualization tools.
Response 5: Thank you for this important recommendation. To address this, we have added a new table, Table 1, in the subsection 3.2, Discussion, that provides a comparative analysis and key attributes of our system versus existing EMR systems and visualization platforms, such as 2D interactive visual electronic medical record system (Sekimoto et al., 2020), and AnamneVis (Zhang et al. 2011).
Round 2
Reviewer 1 Report
Comments and Suggestions for Authors
Dear Authors,
Thank you for submitting the revised version of your manuscript and for carefully addressing all the comments and suggestions.
I am pleased to confirm that I accept the manuscript in its current form.
With best regards,
Reviewer
Reviewer 2 Report
Comments and Suggestions for Authors
The authors addressed all my comments. I suggest the acceptance of the paper.